# Correlations between Contrast-Enhanced Ultrasound Imaging and Histopathological Results in Salivary Gland Lesions

**DOI:** 10.3390/diagnostics12112636

**Published:** 2022-10-31

**Authors:** Karolina Krupa, Patryk Wieczorek, Olga Scrinscaia, Anna Puzio, Zbigniew Kozłowski, Wioletta Pietruszewska, Ludomir Stefańczyk

**Affiliations:** 1Department of Otolaryngology, Head and Neck Oncology, Norbert Barlicki Memorial Teaching Hospital No. 1, Medical University of Lodz, 90-153 Lodz, Poland; 2Department of Radiology and Diagnostic Imaging, Norbert Barlicki Memorial Teaching Hospital No. 1, Medical University of Lodz, 90-153 Lodz, Poland; 3Department of Pathology, Medical University of Lodz, 92-213 Lodz, Poland

**Keywords:** CEUS, salivary gland lesion, histopathological results

## Abstract

The purpose of this study was to search for correlations between contrast-enhanced ultrasound (CEUS) imaging and histopathological results in salivary gland lesions and to determine the accuracy of CEUS in the preoperative differentiation of salivary gland tumours according to postoperative histopathological results. The study included 54 consecutive patients with 63 salivary gland lesions who underwent CEUS examination prior to surgical treatment at the Department of Otolaryngology, Medical University of Łódź (Poland) in 2019–2022. The accuracy of CEUS in differential diagnostics of salivary gland lesions was later verified against final histological diagnosis. Among 63 salivary gland lesions, 26 were categorized as malignant or with malignant potential, and 37 were benign. There was a correlation between professional photographs of CEUS imaging and microscope slides containing postoperative specimens. A strong heterogeneous enhancement was observed mainly in benign lesions, with while weak heterogeneity mostly among the malignant or with malignant potential lesions. A pattern of contrast enhancement in specific structures reflected histopathological images. These results suggest that contrast-enhanced ultrasonography is a promising tool for the preoperative diagnostics of salivary gland lesions.

## 1. Introduction

There are many types of salivary gland tumours, including those caused by inflammatory or autoimmunologic processes. “Up to 80% of salivary gland neoplasms occur in the parotid gland” [1]. “The most common histologic types of benign parotid tumours are pleomorphic adenoma and Warthin tumour, while the most often diagnosed malignancies of the parotid are mucoepidermoid carcinoma and adenoid cystic carcinoma” [2]. While Warthin tumour is untroublesome to differ from parotid malignancies, differential diagnostics between malignant tumour and pleomorphic adenoma in some cases remains a challenge. Preoperative diagnostics should provide crucial information on salivary gland lesions, among which the type of pathology is determinant as far as resection method is concerned. Other features, such as tumour’s size, topography, infiltration of other structures and vascularisation of both adjacent structures and the tumour itself may well facilitate planning the surgery. Fine-needle biopsy is frequently used for the preoperative differentiation of salivary gland tumours, but owing to sampling difficulties and the great heterogeneity of these neoplasms (especially pleomorphic adenoma), the results are not always conclusive. Moreover, there might always be a risk of spreading neoplasmatic cells during the procedure. Therefore, in the cases of patients with concomitant diseases, for whom a surgery may be hazardous, it is crucial that decisions of watchful waiting be based on safe and reliable diagnostics techniques. “Fine needle biopsy should not be relied upon as the sole determinant of a surgeon’s management plan due to its limited sensitivity” [3]. Methods currently used for differential diagnostics of salivary gland tumours include conventional ultrasound, computed tomography (CT) and magnetic resonance imaging (MRI). A contrast-enhanced USG might be more accurate than conventional USG. It might also pose a cheaper, faster and of comparable accuracy alternative to CT or MRI. “CEUS has the potential to differentiate salivary gland lesions preoperatively” [4]. It appears to be a promising diagnostic tool, especially in combination with clinical complaint assessment [4,5]. Guiban et al. in 2021 suggested that “the combination of B-mode US and CEUS greatly improved the sensitivity of the CEUS performed individually and presented remarkable accuracy, with the potential to reduce the number of invasive procedures” [6]. Saito et al. in 2020 reported that “CEUS is a simple, safe, and noninvasive examination for patients that can be performed conveniently in real-time and could help differentiate PMA from WT in salivary glands” [7]. Papers present in the literature suggest that there is a constant need for further studies with large patient groups [4,5,6,7]. The authors had access to a relatively large group of patients who underwent preoperative contrast-enhanced ultrasound (CEUS) examination. The aim of this study was to search for correlations between CEUS imaging and the results of postoperative histopathological examinations of surgical specimens and to establish the accuracy of CEUS in differential diagnostics of salivary gland lesions.

## 2. Materials and Methods

A prospective study was conducted in the Otolaryngology Department and Radiology Department, Medical University of Łódź. The study included 54 consecutive patients with 63 salivary gland lesions treated surgically at the department in 2019–2022. The group included 21 men and 33 women 21 to 82 years old. The study was conducted according to the guidelines of the Declaration of Helsinki and approved by the Bioethics Committee at the Medical University of Łódź, and all patients gave their informed consent to participate in the project. Before surgery, conventional USG and CEUS of salivary glands were performed in every patient. The time between the CEUS examination and the surgery did not exceed 10 days, and most often it was less than 48 h. Surgical specimens were subjected to routine histologic examination at the Department of Pathomorphology, Medical University of Łódź. Based on histopathological examinations, the lesions were classified into two groups. The first group involved benign lesions: monomorphic adenomas, including Warthin tumours, an oncocytic adenoma, myoepithelial adenomas, a lymphoepithelial cyst, sarcoidosis, lymph nodes, a fibrolipoma and inflammatory lesions. The second group involved malignant or of malignant potential lesions: a lymphoepithelial carcinoma, an adenoid cystic carcinoma, pleomorphic adenomas and basal cell adenomas.

### 2.1. Image Acquisition and Processing

GE logiq 7 Espert system with linear probe 9 L was used. CEUS was performed according to literature guidelines for CEUS in parotid glands [8,9]. Examination included standard grey scale (B-mode) ultrasound. Size, localization and quantity of lesions were noted. Next, colour Doppler was performed. In the last step, CEUS was performed starting from the injection of 4.8 mL of contrast agent (SonoVue, Amsterdam, Netherlands) into the medial cubital vein. CEUS was performed with low mechanical index (<0.1) to avoid the destruction of contrast agent bubbles and the disruption of signal caused by it [10,11]. Three main phases of acquisition were noticed. The acquisition of each phase was intervallic: arterial phase (10–45 s), portal venous phase (30–120 s), late venous phase (120–640 s) [10,12]. During examination, enhancement of tumour in arterial phase was compared with salivary gland parenchyma.

SonoVue is a contrast agent containing sulphur hexafluoride (SF6) microbubbles for CEUS imaging. “It is used to enhance the echogenicity of the blood, which can improve the signal-to-noise ratio in US. (…) Low solubility gas contrast agents such as SonoVue allow imaging at low mechanical index, which in turn leads to effective tissue signal suppression” [13].

### 2.2. Image Analysis

The following factors were taken into consideration: shape (round, multiple curved, blurred), echogenicity (hypoechogenic, isoechogenic, hyperechogenic) and specific pattern of contrast enhancement (strong homogeneous, weak homogeneous, strong heterogeneous, weak heterogeneous). The enhancement level referred to the enhancement of an intact salivary gland tissue (Figure 1, Figure 2 and Figure 3). The additional variable was patient’s age.

Two models were analysed. Only the first one included contrast enhancement. The CEUS images were in the following step compared with professional photographs of microscope slides containing postoperative specimens after preparation and proper staining at the Department of Pathomorphology, Medical University of Łódź. The authors focused on specific contrast enhancement areas and structures revealed on photographs that overlapped.

### 2.3. Statistical Analysis

Statistical analysis was performed in TIBCO Statistica 13.3 Software. (TIBCO, Palo Alto, CA, USA).

Categorical variables were reported in 2 × 2 contingency tables, and chi-squared tests with Yates correction for continuity were applied where total number of cases was smaller than 100 and greater than 5 cases in any analysed cell. Fisher’s exact tests were applied when any cell count was lower than 5, and Haldane-Anscome correction was applied to cell counts lower than 1. For 2 × 3 and 2 × 4 tables, the Freeman-Halton extension was applied. Where applicable, Odds Ratio and 95% Confidence Interval was calculated. All calculations were carried out with the threshold of statistical significance set at a *p* value no greater than 0.05, unless otherwise mentioned.

#### Multivariate Analysis

To perform logistic regression analysis of the probability of correct assignment of a salivary gland lesion to one of the abovementioned groups (benign and malignant/with malignant potential), the following characteristics were analysed: age, shape, echogenicity and pattern of contrast enhancement. As the first step in building a logistic regression model, a univariate analysis was performed using the likelihood ratio (LR) test.

Each variable with *p* value from the LR test no greater than 0.35 was included in the multivariate analysis using mixed-effects logistic regression. Continuous variables were checked for linearity. There were no statistically significant interactions between the variables. 

To assess the influence of contrast enhancement on diagnostic accuracy, two logistic regression models were constructed. Age, shape and contrast enhancement were included in model, while model 2 included only age and shape. The cut-off values for both regression models were established based on receiver operating characteristic (ROC) curve analysis, and the highest Youden’s index determined the proposed cut-off value. To assess the diagnostic performance of the CEUS, the measures of occurrence (sensitivity, specificity and accuracy) and the possibility of discriminating (positive and negative predictive values) were calculated per the determined cut-off values.

## 3. Results

Among the 63 salivary gland lesions, 26 (41.27%) were categorized as malignant or with malignant potential and 37 (58.73%) as benign. Mean age among the whole group was 60.30 years with SD = 12.77 and OR = 0.972 (0.934; 1.013).

In the first group, strong homogeneous enhancement was found for 1 (3.85%) lesion; weak homogeneous, 1 (3.85%); strong heterogeneous, 10 (38.46%) and weak heterogeneous, 14 (53.85%). In the second group, strong homogeneous enhancement was found for 2 (5.41%) lesions; weak homogeneous, 2 (5.41%); strong heterogeneous, 26 (70.27%) and weak heterogeneous, 7 (18.92%). Within the first group, 9 (34.62%) lesions were round, 16 (61.54%) were multicurved and 1 (3.85%) was blurred. Within the second group, 7 (18.92%) lesions were round, 27 (72.97%) were multicurved and 3 (8.11%) were blurred.

A correlation between professional photographs of CEUS imaging and those of microscope slides containing postoperative specimens was observed. It appears that the strong heterogeneous enhancement pattern is more likely to occur in benign lesions, while malignant, and malignant potential lesions tend to present weak heterogeneous enhancement. The foundation of such tendencies is revealed in pathomorphological specimens. Various tissues such as fluid and cartilage commonly present in salivary gland tumours, are generally poorly enhanced. The enhancement of such areas in CEUS is poor if any. The cross-section of Warthin tumour, for example, is presented by one or more often numerous minor cysts containing mucus. 

Those not-enhanced areas and structures revealed on histopathological photographs overlapped (Figure 1, Figure 2, Figure 3, Figure 4, Figure 5 and Figure 6).

Univariate analysis revealed that all lesions except for one were hypoechogenic on conventional US; therefore, echogenicity was not included in the regression model (Table 1). The ultrasound images were assessed by two people according to a scheme by consensus.

In the multivariate analysis, only contrast enhancement was statistically significant. Weak heterogeneous contrast enhancement was 5.2 times more likely to be a malignant or of malignant potential lesion. (OR 5.2 95% CI 1.623; 16.656, *p* = 0.006).

Area under the curve (AUC) was compared between ROC curves prepared for each regression model. The AUC in model 1 with CEUS (Figure 7) was 0.770 (SE 0.062), while in model 2 without CEUS (Figure 8) it was 0.683 (SE 0.0687), a difference of 0.087. The addition of contrast enhancement to the regression model improved specificity by 16.2% while lowering the sensitivity by only 3.9% (Table 2).

## 4. Discussion

This study shows that preoperative contrast-enhanced ultrasonography is a promising tool for the differential diagnostics of salivary gland lesions. The authors searched for factors that might contribute to higher diagnostic accuracy of ultrasound imaging. Analysis showed that there were characteristic features in CEUS imaging and specific patterns of contrast enhancement in tumours’ structures that reflected photographs of pathomorphological preparations to a certain extent. Not only does the study show the vascularisation of a lesion and adjacent structures, but it also presents that specific structures within salivary gland and the tumour itself, such as cicatricial tissue, connective tissue, fluids or necrosis, have individual patterns of size and shape of enhancement after contrast injection. Due to numerous disadvantages of fine-needle biopsy, there is a constant need to develop radiological, noninvasive methods of preoperative differentiation among salivary gland pathologies. The role of MRI in this field is growing significantly. Currently, MRI plays a key role in the preoperative diagnosis of tumours. Should a surgeon consider a watchful waiting attitude in patients suffering from numerous diseases, there is a need for a sensitive, specific, repeatable and noninvasive procedure that could help qualify a lesion as either malignant or prone to malignant transformation. A crucial thing would be to capture the moment of possible turning a benign tumour into a malignant one. In such cases, when a surgery and anaesthesia pose a major risk for the patient, avoiding biopsy with potential neoplasmatic cells spreading around healthy tissue should be taken into consideration. “Carcinoma Ex-Pleomorphic Adenoma (CExPA) accounts for 5% to 15% of all salivary gland malignancies and can arise in up to 25% of untreated pleomorphic adenoma. Malignant transformation is often seen in recurrent pleomorphic adenoma (PA) with the risk of transformation ranging from 5% to 10% for untreated pleomorphic adenomas over a 15-year period” [14]. “Basal cell adenomas follow PA and WT as the third most common benign parotid tumour. Their distinctive membranous subtype (…) has the highest risk of malignant transformation (up to 28%)” [15]. “It is noteworthy that in most cases, in which the biopsy results were falsely negative, the parotid malignancies were misdiagnosed preoperatively as pleomorphic adenomas. According to the literature, because of their heterogeneous histologic structure, distinguishing pleomorphic adenomas from other tumour types, including malignant ones, can be challenging even for an experienced pathologist” [2]. The study findings imply that contrast-enhanced USG would be able to provide additional information helpful for establishing the differential diagnostics of salivary gland tumours. The ultrasound technique provides more detailed information, enabling the diagnosis to be made before the operation. Promising observations concern elastography. The authors are aware of the potential limitations of this study. The present findings suggest that future research should focus on the role of contrast-enhanced USG in distinguishing salivary gland malignancies from specific histologic types of benign lesions, particularly as far as pleomorphic adenoma is concerned. In CEUS imaging, the experience of a radiologist has a major impact on the results and their repeatability. Moreover, the still limited availability of this method might be an obstacle in making CEUS a standard procedure. On the other hand, as far as both patient’s reconvalescence and cost-effectiveness are concerned, a routinely performed preoperative CEUS, even combined with a fine-needle biopsy, might be a cheaper alternative to repeated surgery after the suboptimal resection of a malignant tumour initially misdiagnosed as a benign one. Even considering all potential limitations, the present results suggest that contrast-enhanced USG is a promising tool in the preoperative diagnostics of salivary gland lesions.

## Data Availability

The data presented in this study are available on request from the corresponding author.

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
