# Peer review of "Correlations between Contrast-Enhanced Ultrasound Imaging and Histopathological Results in Salivary Gland Lesions"

_diagnostics, 2022, doi:10.3390/diagnostics12112636_

Round 1

Reviewer 1 Report

The authors should clearly show that the accuracy of CEUS is assured even by different diagnosticians.

Imaging diagnosis is judged by the eyes of the diagnosing physician, and different diagnoses may be made for the same image.

Therefore, it is necessary to clarify whether the CEUS imaging results as shown in Table 1 can be guaranteed even when different diagnosing physicians are involved. 

Author Response

Response 1: We thank the reviewer for pointing this out. We clarified the way the ultrasound images were asssesed in “Results” section.

Reviewer 2 Report

Krupa et al. presented "A correlation between contrast enhanced ultrasound imaging and histopathological results in salivary gland lesions.". In this work a correlation between contrast enhanced ultrasound (CEUS) imaging and histopathological results in salivary gland lesions is studied to study the accuracy of CEUS in preoperative differentiation of salivary gland tumors.  Overall, the manuscript is well written, and the results supports the claims of the authors. I have the following minor comments:

*Introduction should be extended to include more recent works relevant to the study.

*Graphics in figure 7 and 8 should be improved. The axes labels can be enlarged for better readability.

Author Response

Response 2: We thank the reviewer for carefully reading our manuscript and pointing these out. We have made the corrections by enlarging axes labels in figures 7 and 8 and extending the introduction, which caused changes in Bibliography. We also corrected two spelling errors.
